# Orthodontic Treatment Does Not Affect Frontal Sinus Development in Female Adults: A Clinical Study

**DOI:** 10.3390/jcm12030778

**Published:** 2023-01-18

**Authors:** Masaki Sawada, Saya Suetake, Hiroshi Yamada, Masaaki Higashino, Susumu Abe, Eiji Tanaka

**Affiliations:** 1Yamada Orthodontic Office, Izumiotsu, Osaka 595-0025, Japan; 2Department of Otorhinolaryngology, Head and Neck Surgery, Osaka Medical and Pharmaceutical University, Takatsuki, Osaka 569-8686, Japan; 3Department of Comprehensive Dentistry, Tokushima University Graduate School of Biomedical Sciences, Tokushima 770-8504, Japan; 4Department of Orthodontics and Dentofacial Orthopedics, Tokushima University Graduate School of Biomedical Sciences, Tokushima 770-8504, Japan

**Keywords:** paranasal sinus, frontal sinus, orthodontic treatment, computed tomography

## Abstract

Frontal sinus growth is gradual and lasts until post-puberty. However, the influence of biomechanical stimuli, such as orthodontic treatment, on frontal sinus development after the growth period has ended remains unclear. This study was designed to elucidate the impact of orthodontic treatment on the frontal sinus morphology of adult females. Sixty women were included and divided into three groups, based on the Frankfort mandibular plane angle. All participants underwent computed tomography and lateral cephalometry before and after treatment. Although two participants exhibited frontal sinus agenesis, most exhibited a bilaterally symmetric frontal sinus without fusion. The frontal sinus width and height were almost similar, irrespective of the vertical skeletal pattern, where the frontal sinus depth was significantly larger in the average mandibular plane angle group than in the low- and high-angle groups. Furthermore, the sinus volume in the low-angle group was likely smaller than that in the average and high-angle groups. On comparing pre-treatment and post-treatment measurements, minimal or no changes to the frontal sinus dimension were detected after treatment. In conclusion, orthodontic treatment did not affect frontal sinus development after the end of growth.

## 1. Introduction

The paranasal sinuses, including the maxillary sinus, frontal sinus, sphenoid sinus, and ethmoidal sinus, occupy a large volume of the cranium and are critical for the air filtering and immune barrier functions of the nose. The tissue membrane on the inner surface of the sinus is covered by a thin layer of mucus, which helps keep the membrane damp and salubrious [1,2,3,4]. Preuschoft et al. [5] reported that paranasal sinuses require biomechanical stimulation for their development and the maintenance of skull architecture. Masticatory stimulation may be associated with the development of paranasal sinuses as mastication is a major contributor to mechanical stress induction in the craniofacial skeleton [6].

The frontal sinus is the most complex sinus because of its location and anatomical variations. At birth, the frontal sinuses are absent and gradually develop until post-puberty. No age-associated differences in frontal sinus dimensions have been reported during physical growth [7,8], and no further expansion has been observed in males over 16 years of age and females over 14 years of age [9]. Recently, we investigated frontal sinus dimensions using three-dimensional (3D) computed tomography (CT) images, obtained at pre-treatment and post-treatment in female adolescents aged 10.3–17.3 years, and observed that the frontal sinus dimensions were significantly larger following orthodontic treatment, regardless of the maxillomandibular jaw–base relationship [10]. Meanwhile, because these changes were small, increases in the frontal sinus dimensions may be caused by pubertal growth and not orthodontic treatment.

Even after pubertal growth ceases, orthodontic and orthognathic forces may act as biomechanical stimulators for sinus development [11]. Said et al. [6] examined the role of adequate anterior occlusion in frontal sinus development using cephalograms. They found that the anterior occlusion involves frontal sinus development, suggesting that the frontal sinus dimensions might be promising parameters for assessing harmonious anterior occlusion. In other words, a larger frontal sinus may have a critical role, such as that of a bumper, in occlusal force transmission. Meanwhile, Benington et al. [12] reported that patients with an anterior open bite showed a tendency toward a larger frontal sinus, indicating that less or no transmission of occlusal forces along the nasal pillars might be involved in the enlargement of the frontal sinus. These findings suggest that frontal sinus dimensions may serve as useful parameters for the evaluation of post-treatment stability [6].

The stability of human frontal sinus size has been studied extensively [13,14,15]. However, these studies have shown two-dimensional (2D) data of the frontal sinus, based on simple radiography and CT, and little information is available on frontal sinus dimensions in humans. Therefore, we aimed to identify the frontal sinus morphology in adult females after pubertal growth using CT and to assess the correlation of vertical craniofacial morphology with frontal sinus morphology. Furthermore, the purpose of this study was to clarify the influence of orthodontic treatment on frontal sinus development in postpubertal female patients. 

## 2. Materials and Methods

This study was designed as a longitudinal clinical study, following the STROBE checklist [16].

### 2.1. Setting

Patients were recruited from the Yamada Orthodontic Office between April 2012 and July 2022.

### 2.2. Participants

The inclusion criteria for enrollment were an age > 18 years and a diagnosis of malocclusion that was treated with conventional orthodontics. Informed consent was obtained from each participant. The exclusion criteria were the presence of paranasal sinusitis, rhinopharyngeal diseases, hormonal disorders, congenital and acquired abnormalities in the craniomaxillofacial region, and a history of orthodontic or orthognathic treatment. The research procedure and protocol were approved by the Ethical Committee at Tokushima University Hospital (approval no. 3900). 

### 2.3. Variables and Data Sources

For each patient, pre-treatment and post-treatment CT were performed with a CT system (Alphard-3030, Asahi Roentgen Ind. Co., Ltd., Kyoto, Japan) under the following roentgenographic conditions: 60–110 kV; 3–15 mA; collimation, 0.6 mm; rotation time, 18 s; lamination thickness, 0.39 mm. A series of images was treated using Dolphin Imaging (Dolphin Imaging & Management Solutions, Verona, Italy) for orthodontic diagnosis and treatment planning. Contiguous CT images were compiled, 3D frontal sinus models were constructed, and volume-rendered images were extracted. Furthermore, lateral cephalometry was performed at the pre-treatment and post-treatment stages using a roentgenographic system (Hyper-X CM, Asahi Roentgen Ind. Co., Ltd.). All radiographs were performed in the standing position with maximum intercuspation. 

Cephalometric tracing for all patients was performed on tracing paper by one examiner (M.S.). Nineteen landmarks were plotted using Dolphin Imaging by another examiner (S.S.) to measure craniofacial morphology. The validity and reproducibility of the plots were verified by other orthodontic experts who participated in this research as collaborators. All examiners were blinded to the general status of the participants. To examine the intra-examiner reliability of cephalometric measurements, one examiner traced five randomly selected cephalograms before measurement and plotted three arbitrary points (nasion, sella, and pogonion) twice within an interval of one week. The intraclass correlation coefficient (ICC) was calculated using SPSS 27.0 (SPSS Inc., Chicago, IL, USA). The ICC was 0.977 (0.957–0.988: 95% confidence interval), confirming the strong reliability of the selected measurements.

### 2.4. Sample Size

The sample size here was calculated based on a previous study comparing three groups divided according to their vertical skeletal type [10]. Accordingly, the effect size was considered to be 0.23. Statistical power (1-β) was obtained by means of the G*Power software [version 3.1.9.7; available from: https://www.psychologie.hhu.de/arbeitsgruppen/allgemeine-psychologie-und-arbeitspsychologie/gpower (accessed on 28 September 2022)]. For the power analysis, we conducted a two-way repeated-measurement analysis of variance (r-ANOVA) using an effect size of 0.23, a significance level of 0.05, and a power level of 0.8. Power analysis was conducted, and the total estimated sample size was calculated to be 51 upon performing a two-way r-ANOVA as a post hoc test.

### 2.5. Quantitative Variables

#### 2.5.1. Craniofacial Morphology

Based on the Frankfort mandibular plane angle (FMA), which indicates the vertical skeletal discrepancies, the participants were divided into three groups: high-angle group (participants with an FMA ≥ 31.0°), average-angle group (FMA > 24.0° but <31.0°), and low-angle group (FMA ≤ 24.0°). From each cephalogram, 13 angular and 8 linear items were measured to evaluate the morphometry of the anterior cranial base, maxilla, and mandible, as shown in Figure 1. The angular and linear measurement items are defined as follows.

Angular measurement items:SNA angle: anteroposterior maxillary position relative to the anterior cranial base.SNB angle: anteroposterior mandibular position relative to the anterior cranial base.ANB angle: the anteroposterior relationship between the maxilla and mandible.Facial angle: chin prominence relative to the Frankfort horizontal plane.Y-axis: angle between the sella-gnathion line and Frankfort horizontal plane.Gonial angle: angle between the mandibular and ramus planes.FMA: divergence of the mandibular plane relative to the Frankfort horizontal plane.Occlusal plane angle: angle between the occlusal plane and the sella and nasion line.Palatal plane angle: angle between the anterior and posterior nasal spine line and Frankfort horizontal plane.U1 to SN: the labiolingual inclination of the maxillary central incisors relative to the anterior cranial base.Interincisal angle: angle between the long axes of the maxillary and mandibular central incisors.IMPA: labiolingual inclination of the mandibular central incisors relative to the mandibular plane.FMIA: labiolingual inclination of the mandibular central incisors relative to the Frankfort horizontal plane.

Linear measurement items:SN: anteroposterior length of the anterior cranial base.Overbite: a vertical gap between the maxillary and mandibular central incisal edges along a line perpendicular to the occlusal plane.Overjet: the anteroposterior gap between the maxillary and mandibular central incisal edges along the occlusal plane.Wits appraisal: the anteroposterior distance between the lines extending perpendicular to the occlusal plane from points A and B.N-Me: the distance between the nasion and menton, indicating anterior facial height.Ar-Go: the distance between the articulare and gonion, indicating mandibular ramus height.Ar-Me: the distance between the articulare and menton, indicating effective mandibular length.Go-Me: the distance between the gonion and menton, indicating mandibular body length.

#### 2.5.2. Frontal Sinus Morphology

The 3D frontal sinus models, obtained from CT images, were analyzed to measure their maximum dimensions. The frontal sinus volume was defined as the total space of the air-filled cavity in the frontal bone, and volume-rendering images were used to automatically calculate the frontal sinus volume (Figure 2). The maximum width was defined as the distance between the right- and left-side points of the frontal sinus. The maximum height was defined as the distance between the baseline and the highest points. The maximum depth was measured as the distance between the most prominent front and rear points of the frontal sinus. Morphological characteristics were determined by appearance, whether bilateral or unilateral, symmetrical or asymmetrical, and fusion or separation. According to previous studies [17,18], the frontal sinus shape can be classified into three types: fan-shaped, quadrangular, and irregular.

### 2.6. Statistical Methods

We used SPSS 27.0 for the statistical analyses in this study. The average sinus dimensions and volume at pre-treatment and post-treatment were calculated for each FMA-based group. The normality test was performed using the Shapiro–Wilk test for each variable regarding data on treatment time, sinus dimensions, and volume for group comparisons; these data showed normal distributions. The frontal sinus features were verified by group analysis using the chi-square test. A statistical analysis of treatment time was performed using a general linear model (GLM) analysis for the three groups, and Student’s *t*-test with a Bonferroni correction was adopted as a post hoc test. Furthermore, a GLM analysis for repeated measures was conducted to compare differences in the sinus dimensions and volume at pre-treatment and post-treatment and among the three groups. For intergroup comparisons, a post hoc test was conducted using a paired *t*-test with the Bonferroni method. The standardized coefficients, which were calculated by the analysis of a linear single-regression test, generally refer to the standard partial regression coefficient and show its influence. This value was used to detect correlations between the frontal sinus morphology and cephalometric measurements in each group. This study considered a probability below 0.05 as the statistically significant value for a type-I error (α).

## 3. Results

### 3.1. Participants

Among the participants, two patients with a low mandibular plane angle showed frontal sinus agenesis and were therefore excluded. The final sample size of 58 patients was adequate for statistical analyses. The participants were divided into three groups before orthodontic treatment, according to FMA: low-angle group (18 women; mean age, 23.2 ± 3.9 years); average-angle group (20 women; mean age, 23.3 ± 3.4 years); and high-angle group (20 women; mean age, 23.0 ± 3.4 years). There was no significant difference in age between the three groups (*p* = 0.962). The treatment times were 3.1 ± 1.7, 3.5 ± 1.1, and 3.9 ± 2.6 years for the low, average, and high-angle groups, respectively. The treatment time was significantly longer in the high-angle group than in the remaining two groups (*p* < 0.05).

### 3.2. Volumetric and Dimensional Measurements of the Frontal Sinus

Most participants revealed a bilaterally symmetric frontal sinus without fusion, irrespective of the maxillomandibular jaw–base vertical relationship (Table 1). Regarding the frontal sinus shape, 41 patients (70.7%) exhibited a fan shape, followed by irregular (20.7%) and quadrangular (8.6%) shapes. Frontal sinus features were not significantly affected by the vertical skeletal pattern (*p* = 0.325), indicating that there were no specific characteristics in the frontal sinus. The frontal sinus exhibited minimal or no changes in morphologic characteristics after orthodontic treatment.

The width, height, and depth of the frontal sinus before orthodontic treatment were 45.1 ± 12.5 mm, 27.8 ± 8.0 mm, and 19.6 ± 4.1 mm in the low-angle group; 47.5 ± 14.6 mm, 29.9 ± 5.9 mm, and 23.0 ± 4.9 mm in the average-angle group; and 48.9 ± 13.3 mm, 31.1 ± 5.1 mm, and 20.2 ± 4.5 mm in the high-angle group, respectively (Table 2). The frontal sinus volumes were 4426.3 ± 2419.4 mm^3^, 5914.7 ± 3564.1 mm^3^, and 5324.5 ± 2417.0 mm^3^ in the low, average, and high-angle groups, respectively. No significant differences in frontal sinus volume were found among the three groups; however, the sinus volume in the low mandibular plane angle group was likely to be smaller than that in the average and high-angle groups. Throughout orthodontic treatment, the frontal sinus depth was significantly greater in the average-angle group than in the other two groups (*p* < 0.05).

When comparing the measurements before and after treatment, a minimal or no change in the frontal sinus dimensions was observed after treatment (Table 2). The average mandibular plane angle group showed a slight increase in the volume of the frontal sinus, although the frontal sinus dimensions showed few or no changes.

### 3.3. Association between Craniofacial Morphology and Frontal Sinus Dimension

We performed a simple linear regression analysis to identify the statistically significant differences among cephalometric measurements, treatment time, and skeletal classification (Appendix A). Several measurements had a significant effect on skeletal classification and treatment time. A few significant interactions between the treatment time and skeletal classification were observed for several measurement items, including the SNA, ANB, gonial, and interincisal angles and overbite.

Furthermore, we analyzed the correlations between cephalometric measurements and frontal sinus dimensions (Table 3). For all patients, the SN, the anteroposterior length of the anterior cranial base, had significant positive correlations with sinus height, depth, and volume before treatment and with sinus height and depth after treatment (*p* < 0.05). Furthermore, the frontal sinus height showed significant negative correlations with the SNA, SNB, and facial angles before treatment and with the SNA angle after treatment (*p* < 0.05).

In the average-angle group, the frontal sinus depth had significant negative correlations with the values along the Y-axis and Ar-Go before and after treatment (*p* < 0.05). The SNA and ANB angles also showed significant negative correlations with frontal sinus height (*p* < 0.05), while a significant positive correlation was found between the gonial angle and frontal sinus depth (*p* < 0.05).

In the high-angle group, the gonial angle was significantly negatively correlated with the width and volume of the frontal sinus before and after treatment (*p* < 0.05 or *p* < 0.01). Frontal sinus width was significantly correlated with the interincisal angle and IMPA before treatment (*p* < 0.05) and with the Y-axis, overjet, and Go-Me after treatment (*p* < 0.05). Furthermore, the post-treatment frontal sinus volume had significant negative and positive correlations with the facial angle and N-Me, respectively (*p* < 0.05).

For the low-angle group, post-treatment values of the occlusal plane angle exhibited significant negative correlations with frontal sinus dimensions, while the post-treatment values of the Wits appraisal had significant positive correlations (*p* < 0.05 or *p* < 0.01). Furthermore, SN had significant positive correlations with the frontal sinus width before and after treatment (*p* < 0.05). The palatal plane angle also showed a significant negative correlation with the frontal sinus depth after treatment (*p* < 0.05).

## 4. Discussion

Recently, we investigated the width, height, depth, and volume of the frontal sinus in female adolescents [10] and observed that the frontal sinus dimensions enlarged with pubertal growth, regardless of horizontal skeletal discrepancies. In our pilot study, the pre-treatment dimensions of the frontal sinus in adolescent females were 22.7 ± 5.1 mm in depth, 29.8 ± 7.3 mm in height, 45.8 ± 12.3 mm in width, and 5151.6 ± 2711.4 mm^3^ in volume [10]. These measurements, observed in the present study among adult females, are similar to these values and are consistent with those reported in previous studies [13,18]. Considering that the pre-treatment mean age was 23.1 ± 3.5 years, the frontal sinus dimensions in the present study were assumed to already have achieved their greatest dimensions. The frontal sinus dimensions in adult females revealed minimal or no changes after orthodontic treatment, indicating that orthodontic treatment did not affect the dimensions of the frontal sinus after pubertal growth. On the other hand, the frontal sinus size and volume in adolescent females after orthodontic treatment were significantly greater than those before treatment [10], and we had previously proposed that the increase in the frontal sinus dimensions primarily accompanies pubertal growth and is less dependent on orthodontic treatment. The present results indicate that biomechanical stimulation derived from orthodontic treatment may affect frontal sinus development during the growth period, although its development ceases after pubertal growth, which further suggests the potential contribution of early orthodontic treatment to paranasal sinus development.

Among the low, average, and high mandibular plane-angle groups, the low-angle group showed only a slight increase in the frontal sinus volume after orthodontic treatment, although the frontal sinus width, height, and depth showed minimal or no change. Since the frontal sinus volume was considerably, but not significantly, less in the low-angle group than in the other two groups before orthodontic treatment, the low-angle group still showed the smallest sinus volume after orthodontic treatment. However, the influence of vertical skeletal discrepancies on frontal sinus morphology remains controversial. 

It has been hypothesized that the paranasal sinuses function as a “crumple zone”, which has a protective function for the head against traumatic impact [19,20,21]. Several researchers have examined this theory. Celiker et al. [22] elucidated the correlation between frontal sinus dimensions and mortality in patients with cranial injury and indicated that the larger the sinus, the greater the risk of death, leading to the rejection of this hypothesis. However, Cai et al. [23] investigated the correlation between frontal sinus volume and the severity of cerebral insults following cranial trauma and found that the frontal sinus is likely to contribute to mitigating intracranial injury, supporting the “crumple zone” hypothesis. In any case, the frontal sinus might change in response to the biomechanical stimulations that are indispensable for the development and maintenance of the skull structure [5]. Kjær et al. [11] stated that biomechanical stimulation is essential for frontal sinus development, suggesting that medical treatment, such as orthodontic and orthognathic treatment, may affect frontal sinus size as an environmental factor. Several studies have demonstrated a relationship between the mechanical stresses derived from mastication and frontal sinus development [6,24]. In the present study, 11 women with an anterior open bite (negative overbite) received orthodontic treatment for an appropriate proper interincisal relationship with a positive overbite. These patients with negative overbites included one with an average mandibular plane angle case, eight with a high mandibular plane angle, and two with a low mandibular plane angle; furthermore, only three had an increased frontal sinus volume after orthodontic treatment, while the other three and five patients exhibited no changes and a decrease in the frontal sinus volume after orthodontic treatment, respectively. This implies that the establishment of anterior occlusion after growth does not affect the development of the frontal sinus.

Of 58 women, two patients (2.38%) with a low mandibular plane angle showed agenesis of the frontal sinus. The prevalence of bilateral frontal sinus agenesis has been reported as 0.49% among Turkish adults [25] and 8.32% among Iranian adults [26], which is similar to our results, and the figure was significantly higher in specific populations, such as Alaskan Eskimos (25% and 36% among men and women, respectively) and Canadian Eskimos (43% and 40% among men and women, respectively) [27]. Amusa et al. [28] investigated 24 dried skulls of a Nigerian population and revealed bilateral frontal sinus agenesis in half of them. This implies that the frontal sinus may not contribute to an increased risk of fatality. The large discrepancy in regional prevalence may be attributable to differences in the sample size and variations in measurement techniques and equipment. The identification of common factors among individuals with frontal sinus agenesis may help to determine if the frontal sinus has other critical roles in the cranium.

## 5. Conclusions

We have analyzed the standard frontal sinus dimensions among adult females and have shown that orthodontic treatment after the end of the growth period has minimal or no effect on the dimensions and morphological features of the frontal sinus. This implies that orthodontic treatment did not affect the size and volume of the frontal sinus after pubertal growth. 

## Figures and Tables

**Figure 1 jcm-12-00778-f001:**
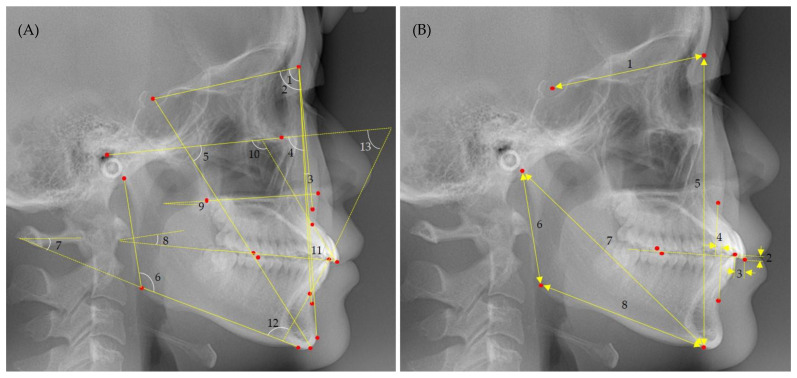
Angular (**A**) and linear (**B**) measurement items for lateral cephalometric analyses. (**A**) 1—SNA angle, 2—SNB angle, 3—ANB angle, 4—facial angle, 5—Y-axis, 6—gonial angle, 7—FMA, 8—occlusal plane angle, 9—palatal plane angle, 10—U1 to SN, 1—interincisal angle, 12—IMPA, 13—FMIA. (**B**) 1—SN, 2—overbite, 3—overjet, 4—Wits appraisal, 5—N-Me, 6—Ar-Go, 7—Ar-Me, 8—Go-Me.

**Figure 2 jcm-12-00778-f002:**
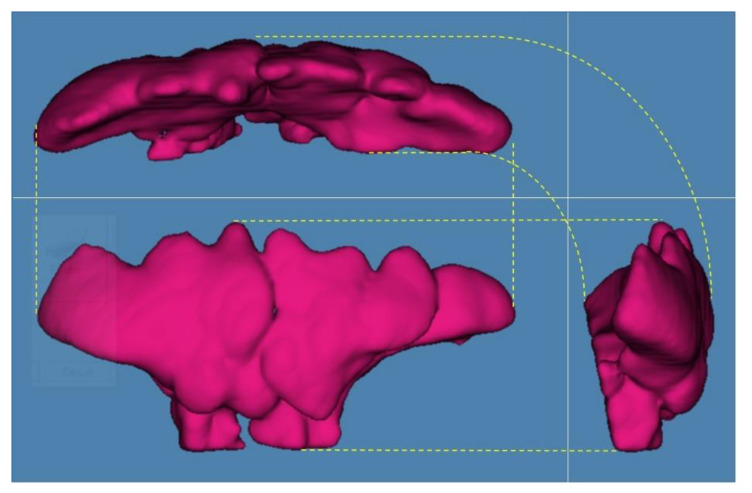
Representative three-dimensional image of the frontal sinus.

**Table 1 jcm-12-00778-t001:** Morphological characteristics of the frontal sinus.

	Low	Average	High	Total	*p*-Value
Bilateral or unilateral					
Bilateral	18	20	20	58 (100%)	
Unilateral	0	0	0	0 (0%)	-
Symmetry or asymmetry					
Symmetry	11	14	16	41 (70.7%)	
Asymmetry	7	6	4	17 (29.3%)	0.441
Spatial relationship					
Fusion	0	2	2	4 (6.9%)	
Separation	18	18	18	54 (93.1%)	0.380
Morphological shape					
Fan-shaped	12	15	14	41 (70.7%)	
Irregular	6	3	3	12 (20.7%)	
Quadrangular	0	2	3	5 (8.6%)	0.325

Low, low mandibular plane angle group; Average, average mandibular plane angle group; High, high mandibular plane angle group).

**Table 2 jcm-12-00778-t002:** Pre-treatment and post-treatment size and volume of the frontal sinus.

		Orthodontic Treatment		*p*-Value		
		Before	After	Interaction	Classification	Time
Age (years)	Average	23.3 ± 3.4	26.8 ± 3.7	0.011	0.973	0.000
	High	23.0 ± 3.4	27.4 ± 3.6			
	Low	23.2 ± 3.9	26.7 ± 4.0			
*p*-value		0.962	0.789			
Width (mm)	Average	47.5 ± 14.6	47.7 ± 14.86	0.551	0.692	0.445
	High	48.9 ± 13.3	48.8 ± 13.1			
	Low	45.1 ± 12.5	45.3 ± 12.6			
*p*-value		0.685	0.692			
Height (mm)	Average	29.9 ± 5.9	29.9 ± 6.0	0.535	0.297	0.267
	High	31.1 ± 5.1	31.6 ± 5.4			
	Low	27.8 ± 8.0	27.7 ± 8.0			
*p*-value		0.285	0.325			
Depth (mm)	Average	23.0 ± 4.9	23.0 ± 5.0	0.427	0.046	0.168
	High	20.2 ± 4.5	20.0 ± 4.3			
	Low	19.6 ± 4.1	19.5 ± 4.1			
*p*-value		0.056	0.038			
Volume (mm^3^)	Average	5914.7 ± 3564.1	5989.5 ± 3685.6	0.276	0.308	0.706
	High	5324.5 ± 2417.0	5200.6 ± 2486.2			
	Low	4426.3 ± 2419.4	4550.6 ± 2521.7			
*p*-value		0.284	0.332			

Red text indicates variables with statistically significant differences between the pre-treatment and post-treatment values. Low, low mandibular plane angle group; Average, average mandibular plane angle group; High, high mandibular plane angle group.

**Table 3 jcm-12-00778-t003:** Correlations between the cephalometric measurements and frontal sinus morphology.

Whole participants																					
Pre-treatment	SNA	SNB	ANB	Facial	Y-axis	Occ.pl.	Go.A	FMA	Pal pl.	U1-SN	IIA	IMPA	FMIA	SN	Overjet	Overbite	Wits	N-Me	Ar-Go	Ar-Me	Go-Me
Width (mm)	ns	ns	ns	ns	ns	ns	ns	ns	ns	ns	ns	Ns	Ns	ns	ns	ns	ns	ns	ns	ns	ns
Height (mm)	*	*	ns	*	ns	ns	ns	ns	ns	ns	ns	Ns	Ns	*	ns	ns	ns	ns	ns	ns	ns
Depth (mm)	ns	ns	ns	ns	ns	ns	ns	ns	ns	ns	ns	Ns	Ns	*	ns	ns	ns	ns	ns	ns	ns
Volume (mm^3^)	ns	ns	ns	ns	ns	ns	ns	ns	ns	ns	ns	Ns	Ns	*	ns	ns	ns	ns	ns	ns	ns
Post-treatment	SNA	SNB	ANB	Facial	Y-axis	Occ.pl.	Go.A	FMA	Pal pl.	U1-SN	IIA	IMPA	FMIA	SN	Overjet	Overbite	Wits	N-Me	Ar-Go	Ar-Me	Go-Me
Width (mm)	ns	ns	ns	ns	ns	ns	ns	ns	ns	ns	ns	ns	ns	ns	ns	ns	ns	ns	ns	ns	ns
Height (mm)	*	ns	ns	ns	ns	ns	ns	ns	ns	ns	ns	ns	ns	*	ns	ns	ns	ns	ns	ns	ns
Depth (mm)	ns	ns	ns	ns	ns	ns	ns	ns	ns	ns	ns	ns	ns	*	ns	ns	ns	ns	ns	ns	ns
Volume (mm^3^)	ns	ns	ns	ns	ns	ns	ns	ns	ns	ns	ns	ns	ns	ns	ns	ns	ns	ns	ns	ns	ns
**Average mandibular plane angle**																					
Pre-treatment	SNA	SNB	ANB	Facial	Y-axis	Occ.pl.	Go.A	FMA	Pal pl.	U1-SN	IIA	IMPA	FMIA	SN	Overjet	Overbite	Wits	N-Me	Ar-Go	Ar-Me	Go-Me
Width (mm)	ns	ns	ns	ns	ns	ns	ns	ns	ns	ns	ns	ns	ns	ns	ns	ns	ns	ns	ns	ns	ns
Height (mm)	ns	ns	ns	ns	ns	ns	ns	ns	ns	ns	ns	ns	ns	ns	ns	ns	ns	ns	ns	ns	ns
Depth (mm)	ns	ns	ns	ns	*	ns	ns	ns	ns	ns	ns	ns	ns	ns	ns	ns	ns	ns	*	ns	ns
Volume (mm^3^)	ns	ns	ns	ns	ns	ns	ns	ns	ns	ns	ns	ns	ns	ns	ns	ns	ns	ns	ns	ns	ns
Post-treatment	SNA	SNB	ANB	Facial	Y-axis	Occ.pl.	Go.A	FMA	Pal pl.	U1-SN	IIA	IMPA	FMIA	SN	Overjet	Overbite	Wits	N-Me	Ar-Go	Ar-Me	Go-Me
Width (mm)	ns	ns	ns	ns	ns	ns	ns	ns	ns	ns	ns	ns	ns	ns	ns	ns	ns	ns	ns	ns	ns
Height (mm)	*	ns	*	ns	ns	ns	ns	ns	ns	ns	ns	ns	ns	ns	ns	ns	ns	ns	ns	ns	ns
Depth (mm)	ns	ns	ns	ns	*	ns	*	ns	ns	ns	ns	ns	ns	ns	ns	ns	ns	ns	*	ns	ns
Volume (mm^3^)	ns	ns	ns	ns	ns	ns	ns	ns	ns	ns	ns	ns	ns	ns	ns	ns	ns	ns	ns	ns	ns
**High mandibular plane angle**																					
Pre-treatment	SNA	SNB	ANB	Facial	Y-axis	Occ.pl.	Go.A	FMA	Pal pl.	U1-SN	IIA	IMPA	FMIA	SN	Overjet	Overbite	Wits	N-Me	Ar-Go	Ar-Me	Go-Me
Width (mm)	ns	ns	ns	ns	ns	ns	**	ns	ns	ns	*	*	ns	ns	ns	ns	ns	ns	ns	ns	ns
Height (mm)	ns	ns	ns	ns	ns	ns	ns	ns	ns	ns	ns	ns	ns	ns	ns	ns	ns	ns	ns	ns	ns
Depth (mm)	ns	ns	ns	ns	ns	ns	ns	ns	ns	ns	ns	ns	ns	ns	ns	ns	ns	ns	ns	ns	ns
Volume (mm^3^)	ns	ns	ns	ns	ns	ns	**	ns	ns	ns	ns	ns	ns	ns	ns	ns	ns	ns	ns	ns	ns
Post-treatment	SNA	SNB	ANB	Facial	Y-axis	Occ.pl.	Go.A	FMA	Pal pl.	U1-SN	IIA	IMPA	FMIA	SN	Overjet	Overbite	Wits	N-Me	Ar-Go	Ar-Me	Go-Me
Width (mm)	ns	ns	ns	ns	*	ns	**	ns	ns	ns	ns	ns	ns	ns	*	ns	ns	ns	ns	ns	*
Height (mm)	ns	ns	ns	ns	ns	ns	ns	ns	ns	ns	ns	ns	ns	ns	ns	ns	ns	ns	ns	ns	ns
Depth (mm)	ns	ns	ns	ns	ns	ns	ns	ns	ns	ns	ns	ns	ns	ns	ns	ns	ns	ns	ns	ns	ns
Volume (mm^3^)	ns	ns	ns	*	**	ns	*	ns	ns	ns	ns	ns	ns	ns	ns	ns	ns	*	ns	ns	ns
**Low mandibular plane angle**																					
Pre-treatment	SNA	SNB	ANB	Facial	Y-axis	Occ.pl.	Go.A	FMA	Pal pl.	U1-SN	IIA	IMPA	FMIA	SN	Overjet	Overbite	Wits	N-Me	Ar-Go	Ar-Me	Go-Me
Width (mm)	ns	ns	ns	ns	ns	ns	ns	ns	ns	ns	ns	ns	ns	*	ns	ns	ns	ns	ns	ns	ns
Height (mm)	ns	ns	ns	ns	ns	ns	ns	ns	ns	ns	ns	ns	ns	ns	ns	ns	ns	ns	ns	ns	ns
Depth (mm)	ns	ns	ns	ns	ns	ns	ns	ns	ns	ns	ns	ns	ns	ns	ns	ns	ns	ns	ns	ns	ns
Volume (mm^3^)	ns	ns	ns	ns	ns	ns	ns	ns	ns	ns	ns	ns	ns	ns	ns	ns	ns	ns	ns	ns	ns
Post-treatment	SNA	SNB	ANB	Facial	Y-axis	Occ.pl.	Go.A	FMA	Pal pl.	U1-SN	IIA	IMPA	FMIA	SN	Overjet	Overbite	Wits	N-Me	Ar-Go	Ar-Me	Go-Me
Width (mm)	ns	ns	ns	ns	ns	**	ns	ns	ns	ns	ns	ns	ns	*	ns	ns	**	ns	ns	ns	ns
Height (mm)	ns	ns	ns	ns	ns	*	ns	ns	ns	ns	ns	ns	ns	ns	ns	ns	*	ns	ns	ns	ns
Depth (mm)	ns	ns	ns	ns	ns	*	ns	ns	*	ns	ns	ns	ns	ns	ns	ns	*	ns	ns	ns	ns
Volume (mm^3^)	ns	ns	ns	ns	ns	**	ns	ns	ns	ns	ns	ns	ns	ns	ns	ns	**	ns	ns	ns	ns

* *p* < 0.05, significantly negative correlation; ** *p* < 0.01, significantly negative correlation; * *p* < 0.05, significantly positive correlation; ** *p* < 0.01, significantly positive correlation.

## Data Availability

Not applicable.

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
