# Peer review of "Orthodontic Treatment Does Not Affect Frontal Sinus Development in Female Adults: A Clinical Study"

_jcm, 2023, doi:10.3390/jcm12030778_

Round 1

Reviewer 1 Report

General comment: We thank the authors for the quality of their work. The topic and the protocols are very interesting. The statistical analyses are well described. Some changes and precisions are needed to improve the quality of the paper. Especially to enable the reader to better understand the differences between this study and those previously published by the authors in 2022.

Comments:

-          It should be indicated in the text before the discussion that the authors have already published an article in 2022 on the same topic. In this previous publication, the subjects were maybe younger and have been separated into three groups according to another skeletal classification. This should be specified clearly.

-          Also it should be precised if the subjects concerned by the first publication are the same than in the present article, as the Ethical authorizations are the same in the two studies.

-          The title should be changed to indicate for example the result and the study type: “Orthodontic treatment does not affect development of the frontal sinus in female adults: a clinical study”.

-          Introduction, l. 65 to 68: the objective of the study is not clear. The word “aimed” is repeated twice. Please make a choice and indicate an objective that fits with the title of the article and that is different in comparison with your previously published articles.

-          Materials and methods: Please respect the STROBE checklist to report observational studies. This implies to change the paragraphs’ titles of the materials and methods. The authors also have to indicate at the beginning of the Materials and Methods part that they have followed the checklist.

-          L71: “Sixty patients”. This must be indicated only in the Results part. Only the sample size calculation must be indicated in Materials and Methods. Moreover, you finally included 58 participants in the analyses so the number “60” is not entirely true.

-          L72: “inclusive” must be replaced by “inclusion”

-          Statistical analysis: very well described. However, you do not confirm that when you have performed Shapiro-Wilk tests, all the quantitative variables have well respected a normal distribution. If it was not the case, have you performed non-parametric tests?

-          Figure 3: is there an interest to conserve this figure as you cite the main values in the text?

-          L203: the word “the” is repeated twice

-          L246: “orthodontic treatment dramatically changed cephalometric measurements”: this is absolutely normal as it is the objective of an orthodontic treatment. Is it important to keep this fact?

-          Discussion, L284 to 289: The first paragraph of the discussion is very similar to those of your previous study. The last sentence “this indicates […] the end of growth” is of very high interest and brings something new. This comment is also true for Conclusion.

-          L296: The references #17 and 18 have already reported measurements before your work. Please indicate why your work is different.

-          More general comment #1: is there any influence on the results of the initial indication for orthodontic treatment? Why don’t specify this indication for the several females included?

-          More general comment #2: why have you chosen only females?

Author Response

Reviewer 1

General comment: We thank the authors for the quality of their work. The topic and the protocols are very interesting. The statistical analyses are well described. Some changes and precisions are needed to improve the quality of the paper. Especially to enable the reader to better understand the differences between this study and those previously published by the authors in 2022.

Answer: We deeply appreciate your nice wording. Your comments were constructive and we revised the manuscript according to your suggestions as much as possible.

Comments:

-It should be indicated in the text before the discussion that the authors have already published an article in 2022 on the same topic. In this previous publication, the subjects were maybe younger and have been separated into three groups according to another skeletal classification. This should be specified clearly.

Answer: Thank you for your comments. According to the reviewer’s advice, we added two sentences in the Introduction section. (revision: lines 47-51)

-Also it should be precised if the subjects concerned by the first publication are the same than in the present article, as the Ethical authorizations are the same in the two studies.

Answer: Thank you for your comments. The subjects in this study were different from those in the first study published in 2022. We did not use the same subjects as the first study at all. (no revision)

- The title should be changed to indicate for example the result and the study type: “Orthodontic treatment does not affect development of the frontal sinus in female adults: a clinical study”.

Answer: Thank you for your advice. We changed the title to “Orthodontic treatment does not affect frontal sinus development in female adults: a clinical study” according to your advice. (revision: title)

-Introduction, l. 65 to 68: the objective of the study is not clear. The word “aimed” is repeated twice. Please make a choice and indicate an objective that fits with the title of the article and that is different in comparison with your previously published articles.

Answer: Thank you for pointing it out. We revised the sentences as follows:

Therefore, we aimed to identify the frontal sinus morphology in adult females after pubertal growth using CT and to assess the correlation of vertical craniofacial morphology with frontal sinus morphology. Furthermore, the purpose of this study was to clarify the influence of orthodontic treatment on frontal sinus development in postpubertal female patients. (revision: lines 67-71)

-Materials and methods: Please respect the STROBE checklist to report observational studies. This implies to change the paragraphs’ titles of the materials and methods. The authors also have to indicate at the beginning of the Materials and Methods part that they have followed the checklist.

Answer: Thank you for your important comments. The reviewer is correct, and the STROBE checklist is a very good item for the readers to better understand our clinical study. We revised the Materials and methods section according to the STROBE checklist. (revision: the Materials and methods section)

-L71: “Sixty patients”. This must be indicated only in the Results part. Only the sample size calculation must be indicated in Materials and Methods. Moreover, you finally included 58 participants in the analyses so the number “60” is not entirely true.

Answer: Thank you for your comments. The reviewer is correct. Then, we deleted “sixty” from the Materials and methods and the Results sections. (revisions: lines 77 and 203)

-L72: “inclusive” must be replaced by “inclusion”

Answer: We have done. (revision: line 81)

-Statistical analysis: very well described. However, you do not confirm that when you have performed Shapiro-Wilk tests, all the quantitative variables have well respected a normal distribution. If it was not the case, have you performed non-parametric tests?

Answer: Thank you for your comments. We dealt with the data of treatment time, sinus dimensions, and volume for group comparisons. These data showed normal distribution. Therefore, the non-parametric test was not performed in this study. We wrote this idea in the statistical methods section. (no revision)

-Figure 3: is there an interest to conserve this figure as you cite the main values in the text?

Answer: No, there is. Then, we deleted Figure 3.

-L203: the word “the” is repeated twice

Answer: We deleted “the”.(revision: line 217)

-L246: “orthodontic treatment dramatically changed cephalometric measurements”: this is absolutely normal as it is the objective of an orthodontic treatment. Is it important to keep this fact?

Answer: Thank you for your comment. According to the reviewer’s suggestion, we deleted this sentence. (revision: line 256)

-Discussion, L284 to 289: The first paragraph of the discussion is very similar to those of your previous study. The last sentence “this indicates […] the end of growth” is of very high interest and brings something new. This comment is also true for Conclusion.

Answer: Thank you for your suggestion. We deleted the first paragraph of the Discussion section. (revision: line 299)

-L296: The references #17 and 18 have already reported measurements before your work. Please indicate why your work is different.

Answer: Thank you for your suggestion. There is no difference in measurements of frontal sinus dimensions between our study and previous studies. However, our study was performed longitudinally. (no revision)

-More general comment #1: is there any influence on the results of the initial indication for orthodontic treatment? Why don’t specify this indication for the several females included?

Answer: No, there might not be any influence on the results of the initial indication for orthodontic treatment. In the inclusion criteria, we described the participants have been diagnosed with malocclusion that was treated with conventional orthodontics but not orthognathic surgery. (no revision)

-More general comment #2: why have you chosen only females?

Answer: As you may know, many females have received orthodontic treatment compared to males. Then, it is relatively easy for us to obtain enough samples from female adults. (no revision)

Reviewer 2 Report

It was my pleasure to review the manuscript entitled: “Does Orthodontic Treatment Affect Development of the Frontal 2 Sinus in Female Adults? (ID jcm-2112424)” submitted to the Journal of Clinical Medicine for further evaluation. This study aimed to evaluate the differences in frontal sinus dimensions of adult females in relation to variations in Frankfort mandibular plane angle (FMA) before and after orthodontic therapy. The study was well-written and presented but with minimal scientific value. There were minor English typos such as line #174 (‘correlation’ should be correction) and line #203 (‘the’ were written in double).

Lines 28-30 and lines 357-359; the conclusion should only reflect the current study’s results, which showed no changes in frontal sinus dimensions according to FMA or orthodontic therapy. The fact that a previous pilot study of growing females showed a change in frontal sinus dimension that is mainly due to growth not orthodontic therapy; should not be intercepted that orthodontic therapy could have an influence on frontal sinus growth, nor it should be written in the current study conclusions. Kindly, revise the conclusions in the abstract and in the main text.

The statistical analysis section (lines 169-183) should be revise and written more clearly.

Author Response

Reviewer 2

It was my pleasure to review the manuscript entitled: “Does Orthodontic Treatment Affect Development of the Frontal 2 Sinus in Female Adults? (ID jcm-2112424)” submitted to the Journal of Clinical Medicine for further evaluation. This study aimed to evaluate the differences in frontal sinus dimensions of adult females in relation to variations in Frankfort mandibular plane angle (FMA) before and after orthodontic therapy. The study was well-written and presented but with minimal scientific value. There were minor English typos such as line #174 (‘correlation’ should be correction) and line #203 (‘the’ were written in double).

Answer: Thank you for your nice wording. Also, we thank you found English typos. We revised them. (revisions: lines 191 and 217)

Lines 28-30 and lines 357-359; the conclusion should only reflect the current study’s results, which showed no changes in frontal sinus dimensions according to FMA or orthodontic therapy. The fact that a previous pilot study of growing females showed a change in frontal sinus dimension that is mainly due to growth not orthodontic therapy; should not be intercepted that orthodontic therapy could have an influence on frontal sinus growth, nor it should be written in the current study conclusions. Kindly, revise the conclusions in the abstract and in the main text.

Answer: Thank you for your important comments. We deleted the last sentences associated with the previous study results in the Abstract and Conclusions sections. (revision: lines 27 and 365)

The statistical analysis section (lines 169-183) should be revise and written more clearly.

Answer: Thank you for your comment. We used three kinds of statistical analysis methods in this study. Especially, the Shapiro-Wilk test was performed for the normality test about dealing with the treatment time, the sinus dimensions, and volume. We did not write the statistical analysis methods for non-normal distribution. The lack of this method was confusing for Reviewer. We briefly showed the results of the Shapiro-Wilk test in the statistical methods section. We hope that the text enhances the Reviewer’s and readers’ understanding. (revision: lines 184-188)